# WBSan: Webassembly Bug Detection for Sanitization and Binary-Only Fuzzing

## Abstract

With the advancement of WebAssembly, abbreviated as Wasm, various memory bugs and undefined behaviors have emerged, leading to security issues and discrepancies that affect usability and portability. Existing methods struggle to detect these problems in Wasm binaries due to challenges associated with binary instrumentation and the difficulty of defining legal memory bounds. While sanitizers combined with fuzzing are recognized as effective means for identifying memory bugs and undefined behaviors, current Wasm sanitizers necessitate compile-time instrumentation, rendering them unsuitable for practical scenarios where only binaries are accessible. In this paper, we propose WBSan, the first Wasm binary sanitizer by employing static analysis and Wasm binary instrumentation to detect memory bugs and undefined behaviors. We develop distinct instrumentation patterns tailored for each type of memory issue and introduce *Wasm shadow memory* to address complex memory bugs. Our results reveal that WBSan achieves a 16.8% false detection rate, outperforming current Wasm binary checkers and native sanitizers in detecting memory bugs and undefined behaviors. Furthermore, when compared with the binary-only fuzzer, WBSan uncovers more crashes (1,174 vs. 556) and achieves greater code coverage (162,385 vs. 22,237 unique search paths).

## 1 Introduction

WebAssembly (abbreviated Wasm) [30] is a binary instruction format that serves as a compilation target for high-level languages including C, C++, C#, and more [45]. Wasm can be deployed to the web and other platforms, such as IoTs [39, 40], smart contracts [22, 57, 61], and cloud computing [35, 50, 58].

Analyzing closed-source Wasm binaries is currently not an easy task since such analysis relies on automatic bug detection methods which typically integrate sanitizers [25]. Current approaches for identifying issues in Wasm binaries often utilize fuzzing without adequate sanitization to trigger diverse bug types [33, 38], or are limited to unsound statically targeting only specific categories of bugs [19, 20]. Given the proven effectiveness of sanitizers in conjunction with fuzzing for detecting a wide range of memory errors and undefined behaviors, it is a logical progression to develop sanitizers specifically tailored for Wasm binaries. Sanitizers generally capture the program execution state through static or dynamic instrumentation to conduct relevant checks. Although a prevalent WebAssembly compiler Emscripten already supports ASAN, *Address Santizer*, and UBSAN, *Undefined Behavior Santizer*, in Wasm programs [1], they necessitate instrumentation at the source code or intermediate representation (IR) level during compilation, making it hard to apply them on binary directly. To this end, developing a comprehensive, cost-effective, and automation-integratable binary Wasm sanitizer to detect these issues becomes meaningful.

Implementing such Wasm binary sanitizer presents several challenges: 1) *Different program architectures*. Although substantial work exists on binary sanitizers in the native programming domain [15, 25, 29, 48, 53], Wasm diverges from traditional von Neumann architecture, where code and data are stored together [36]. In Wasm, the code and data are stored separately. Native programs use absolute or relative addressing instructions, along with register-based addressing for jumps or function calls. In contrast, different instruction set of Wasm employs indexing for memory accesses and function calls, complicating the direct application of existing binary static or dynamic instrumentation techniques [59]. Additionally, the current binary rewriting techniques [23, 25] predominantly focus on specific instruction sets and lack a universally applicable framework across different architectures. 2) *Detect legal memory objects*. Extracting the valid memory ranges from Wasm binaries is also a non-trivial task. This complexity arises from the loss of certain memory-related information during the compilation process [18] and the unique memory structure of Wasm, which employs a managed stack for handling specific variables [36].

In this paper, we propose WBSan, a **W**asm **B**inary **San**itizer designed to detect memory bugs and undefined behaviors, which can be effectively integrated with existing binary-only Wasm fuzzers. To enhance performance and reduce false positives, we identify potentially problematic instructions (*anchor instructions*) through the analysis of control and data dependencies. We implement *Wasm shadow memory* in the target binary to more accurately define valid memory boundaries. We use Wasm binary instrumentation [37] to insert specialized dynamic checking patterns without affecting the functionality of the Wasm binary. We create distinct detection patterns for the four types of memory error and six types of undefined behavior in Wasm, statically instrumenting the target binary with these patterns. The instrumented detection patterns can identify and locate erroneous instructions and call traces during execution without disrupting the stack balance of Wasm.

We implement a prototype of WBSan and evaluate its bug detection effectiveness, performance overhead, and adaptability to the current binary-only Wasm fuzzer. We compare bug-finding capability of WBSan with two state-of-the-art Wasm binary checkers, Wasmati [20] and fuzzm-canary [38], as well as three advanced native binary sanitizers, Valgrind [48], QASan [29], and Retrowrite [25]. Although WBSan does not utilize source code information, we also compare it with source-level Wasm sanitizer (ASAN, UBSAN) provided by Emscripten. WBSan demonstrates superior accuracy in detecting memory errors and undefined behaviors in Wasm binaries, outperforming nearly all existing native binary checkers. The instrumentation of WBSan results in an acceptable increase in binary size, with the additional overhead for real large projects being on par with the runtime overhead associated with compiler-based instrumentation. We assess the applicability of WBSan with a binary-only Wasm fuzzer, fuzzm, across 11 real-world programs over 120 hours fuzzing. WBSan detects more crashes and achieves higher code coverage compared to the current fuzzing method.

WBSan achieve superior sanitization effects compared to the current Wasm and native sanitizers. It can be integrated into current fuzzing frame work for fuzzing Wasm binaries, offering strong performance and compatibility, and can serve as a viable drop-in replacements for source-based tools. In summary, this paper makes the following contributions:

- We propose WBSan, the first Wasm binary sanitizer that leverages static analysis and binary instrumentation to detect memory bugs and undefined behaviors.
- We design unique detection patterns for six undefined behaviors and four memory bugs frequently occur in Wasm. These patterns can be easily instrumented in target binary and triggered runtime.
- We introduce *Wasm shadow memory*, which dynamically verifies memory validity by inserting red zones around allocated memory and hooking relevant memory access instructions. *Wasm shadow memory* addresses the challenge of identifying valid memory boundaries within Wasm binaries.
- We evaluate WBSan regarding its bug detection capabilities, performance overhead, and applicability for Wasm binary-only fuzzing, yielding the following results:
  - WBSan achieves a higher bug detection rate across 10 CWEs in 14,278 test samples, surpassing all of the current Wasm binary checkers and native binary sanitizers.
  - WBSan can nearly match the detection rate of source-level sanitizers without source code, and performs better on certain types of bugs (*float-cast-overflow* and *memory-leak*).
  - WBSan introduces acceptable binary size increase and runtime overhead comparable to compiler-based instrumentation.
  - WBSan finds more crashes (1,174 vs. 556) and higher code coverage (162,385 vs. 22,237 unique search paths) across 11 real-world programs compared to existing Wasm binary fuzzing framework.
- We make WBSan publicly available [2].

## 2 Background and Motivation

In this section, we introduce the compilation pipeline of Wasm, the role of sanitizers and their current development status in Wasm. We highlight the shortcomings in the existing research field and explain why there is a need for a binary-only sanitizer.

### 2.1 Webassmebly Binary

WebAssembly binaries are created by compiling from source code in a high-level programming language, such as C or C++, with a compiler. For example, Emscripten [60] is a prevalent C/C++-to-Wasm compiler, which leverages Binaryen [10] and LLVM [13] tools such as Clang [11] internally to generate a Wasm binary. Unlike native x86/x64 binaries, Wasm binaries manage basic variable types (integers, floating-point numbers) through a *managed stack* maintained by the host sandbox, while more complex data types (arrays, objects, etc.) are handled via an *unmanaged stack* [36]. Additionally, memory access Wasm instructions and function calls use indices [36] rather than relative or absolute addresses. The different program structure results in a significantly distinct analysis approach for Wasm compared to traditional program analysis.

**Table 1: Current representative sanitizers in the Native and Wasm binary domains.**

| Instrument Level | Native Binary | Wasm Binary |
|---|---|---|
| Source/IR | ASan [24], KAsan [3], UBSan[15], TySan [26], ... | ASan, UBSan (*Emscripten* [60]) |
| Binary | QASan [29], RetroWrite [25], Valgrind [48], ... | WBSan |

### 2.2 Binary Sanitizer

Binary sanitizers have proven to be an effective way of detecting memory errors and undefined behavior in binaries without source code. They can be used independently for program analysis [25] or combined with fuzzing to automate bug detection process [23]. In recent years, there have been binary-only sanitizers targeting memory errors and undefined behaviors in the native binary domain, both in user space [23, 25, 47] and the kernel [24, 43]. In the Wasm domain, to our knowledge, Emscripten, as an LLVM-based Wasm compiler, currently supports the integration of existing two sanitizers [1], namely ASan and UBSan, during the compilation process at the source code (UBSan) or generated LLVM-IR (ASan). Both of these sanitizers require the source code of the program.

### 2.3 Why a binary-only Wasm Sanitizer

In applications where Wasm is utilized for smart contracts [22, 57, 61], IoT [39, 40], or in browsers [14], users often interact with compiled third-party Wasm modules without access to the source code. Daniel et al. [36] demonstrated that compiling a vulnerable third-party C library into Wasm could facilitate malicious cross-site scripting attacks in browsers. The absence of source code, combined with the fact that such vulnerabilities are typically triggered by specific inputs, complicates their analysis and detection by current Wasm checkers. Existing research has developed fuzzing [31, 38] and symbolic execution [32, 33, 44] frameworks for Wasm binaries; however, not all bugs manifest as crashes or hangs, making detection challenging. Table 1 outlines the current landscape of binary sanitizer for both native and Wasm binaries. To address this gap, we propose WBSan, which assists developers in more effectively detecting memory errors and undefined behaviors in Wasm binaries. WBSan can be integrated seamlessly with existing Wasm fuzzing frameworks to trigger a broader range of bugs.

## 3 Design

This section details our approach to identifying memory bugs and undefined behaviors in Wasm binaries. In section 3.1, we give an overview of WBSan. Section 3.2 introduces WBSan, our binary Wasm sanitizer, and describes its implementation details.

### 3.1 Overview of WBSan

We design WBSan, which effectively performs memory and undefined behavior sanitizing through Wasm bianry instrumentation for the target Wasm binary. The key accomplishments of WBSan are as follows: (1) WBSan provides unique detection patterns for

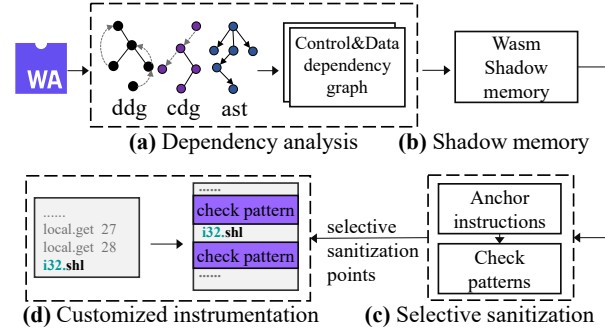

**(a)** Dependency analysis    **(b)** Shadow memory

**(d)** Customized instrumentation    **(c)** Selective sanitization

**Figure 1: Workflow of WBSan**

the four common memory bugs and six undefined behaviors in Wasm, enabling the dynamic acquisition of program internal states and real-time error detection. (2) WBSan introduces *Wasm shadow memory*, which dynamically captures valid memory boundaries by inserting red zones in allocated heaps and stacks, along with hooking relevant access instructions.

Figure 1 presents an overview workflow of WBSan, which consists four key steps: First, WBSan employs static analysis to analyze the control and data dependency of the given Wasm binary (❶). WBSan then hooks memory allocation and deallocation functions, maintaining a *Wasm shadow memory* by inserting red zones in the heap and stack. This *Wasm shadow memory* dynamically tracks valid memory ranges at runtime and reports errors when illegal memory accessed (❷). Next, WBSan conducts pattern matching analysis on all instructions susceptible to memory errors and undefined behaviors (*anchor instructions*). If an anchor instruction is identified, WBSan designates it as a selective sanitization point. (❸). WBSan instruments all selective sanitization points with corresponding detection patterns, while ensuring the target Wasm binary maintains stack balance and type correctness (❹). The instrumented Wasm binary can function independently for sanitization or be integrated with automated detection tools, such as fuzzing, to enhance the detection process.

### 3.2 WBSan Implementation

#### 3.2.1 Control & Data dependency analysis.
The objective of WBSan is to detect memory bugs and undefined behaviors within the target Wasm binaries. These issues are often not apparent in a single instruction; rather, they necessitate specific control or data conditions to be triggered. For instance, a *use-after-free* bug involves three steps: allocating memory, freeing that memory object, and subsequently reusing it. Triggering such a bug requires: 1) The existence of a control flow path from allocation to deallocation to usage; 2) Allocation, deallocation, and usage of the same memory object. To facilitate a more effective analysis of these complex bugs, it is essential to gather sufficient information regarding control and data dependencies within the target Wasm binary.

In this work, we extract and analyze the control dependencies, data dependencies, and function call information from Wasm binaries to: 1) identify the instructions that require instrumentation, and 2) minimize the performance overhead and errors in the instrumented samples. We construct the data dependence graph (DDG), control dependency graph (CDG), and abstract syntax tree (AST)

of Wasm programs to obtain insights into control, data, and function call dependencies. We leverage the C++ APIs provided by the WebAssembly Binary Toolkit [16], which offers a mapping representation from Wasm binaries to C++ objects, enabling straightforward analysis of Wasm functions, memory, and global variables. Our analysis adheres to the standard definitions for DDG, CDG, and AST, with the parsed graphs stored as C++ objects for subsequent analysis. The parsed graphs will be stored in the form of C++ objects and used for subsequent analysis.

#### 3.2.2 Wasm shadow memory.
Identifying valid memory boundaries for memory bugs poses significant challenges due to information loss during the compilation process [18], complicating the detection of such issues. To address this, we introduce *Wasm shadow memory*, which inserts red zones in allocated heap and stack memory objects while hooking relevant memory access instructions to dynamically ascertain valid memory boundaries during the execution of Wasm programs.

While shadow memory is a widely adopted solution in ASan [53], its implementation in Wasm binaries necessitates careful consideration of Wasm's unique memory structure and allocation methods. Notably, the stack of Wasm is different as basic variable type is mantained by the host [36]. Directly inserting red zones and storing shadow memory within Wasm's memory can disrupt the program's original functionality.

WBSan implements red zones by instrumenting memory allocation/deallocation functions. For the stack, in Wasm supporting WebAssembly System Interface (WASI), the stack pointer is the first global variable. For non-WASI Wasm, simple heuristics can be employed to identify the stack pointer [38]. Once identified, we subtract the size of the red zone (defaulting to 16 bytes, configurable) from the stack pointer. At each function exit, the red zone is unpoisoned, and the stack pointer is adjusted by 16 bytes. For the heap allocation and deallocation, such as `malloc`, `calloc`, and `free`. The instrumentation process involves allocating an additional 32 bytes, comprising 16 bytes both before and after the heap region. Upon completion of the allocation function, the preamble and postamble areas are marked as poisoned in the shadow memory. The first four bytes of the preamble store the size of the allocated space, followed by another four bytes containing a magic value used during deallocation. Subsequently, the internal heap region is unpoisoned and made accessible, with the possibility of re-poisoning during deallocation. When the instrumented `store`/`load` instructions access memory, they check whether the target address is accessible in shadow memory to ascertain if illegal memory read or write requests occur.

To align with the memory structure of Wasm, WBSan maps eight bytes of real memory address to one byte of shadow memory, storing this shadow memory in an additional linear memory using the multi-memory mechanism in Wasm [4]. We instrument each memory access to verify whether it attempts to access an invalid address within a redzone. If such an access is detected, an error code is generated, and the program is terminated via an exit function. Algorithm 3 and Algorithm 4 in Appendix show specific redzone instrumentation methods.

Table 3: Matching patterns for undefined behaviors

| Undefined behavior | Anchor instruction(s) | Matching pattern(s) | Constraints |
|---|---|---|---|
| Shift-overflow | i32.shl/i32shr_s/ i32.shr_u/i64.shl/ i64.shr_s/i64.shr_u | ①local.get a ②local.get b ③X | ③→② ③→① |
| Integer-overflow | i32.add/sub/mul/div i64.add/sub/mul/div | ①local.get a ②local.get b ③X | ③→② ③→① |
| | i32.add/i32.sub/ i32.mul/i32.div | ①X ②local.set a ③local.get b ④local.get c ⑤i32.strore16/i32.store8 | ①⇒② ②⇒③ ③⇒④ ②→① ④→② ⑤→④ |
| Implicit-integer-sign-change | i32.load8u/load16u/ i64.load8u/load16u/load32u | ①X ②local.set a | ②→① |
| | $malloc/$dlmalloc/ $memcpy/$memmove/$strncpy | ①X | - |
| Float-cast-overflow | f32.demote_f64 | ①local.get a ②X ③local.set b | ②→① ③→② |
| | i32.trunc_f32/ i32.trunc_f64 | block  ①X  ②local.set a end ③i32.const I32_MIN ④local.set a | ①⇒② ②⇒③ ③⇒④ ②→① ④→③ |
| Null-dereference | i32.load/i64.load | ①local.get a ②local.get b ③X | ③→② ③→① |
| Float-divide-by-zero | f32.div/f64.div | ①local.get a ②local.get b ③X | ③→② ③→① |

**Note: Anchor instruction(s)** indicate potential sink instructions in Wasm. If the anchor instructions for each type of undefined behavior simultaneously match the patterns and corresponding constraints, they can be considered sanitization points for subsequent instrumentation. X in **Matching pattern(s)** represents any specific anchor instruction corresponding to a particular undefined behavior. ②→① indicates that ② has a data dependency on ①, and ②⇒① indicates that there exists a path in control flow from ② to ①.

### 3.2.3 Wasm selective sanitization.

Existing sanitizers primarily define memory errors and undefined behaviors from the perspective of source code. However, in Wasm binaries, there is no direct one-to-one correspondence between binary instructions and lines of code. Since Wasm only supports four basic variable types (i32, i64, f32, f64), more complex variable types, such as arrays and structures, necessitate the use of Wasm unmanaged stack and linear memory for representation and manipulation [36]. Consequently, error points (sink) occurring in the source code cannot be directly mapped to a single binary instruction or variable. Therefore, it is essential to elucidate the specific manifestations of these memory bugs and undefined behaviors at the Wasm binary level to enable effective detection of these issues.

From the perspective of source code, identifying memory errors or undefined behaviors is straightforward, as they typically arise from specific operations on particular variables. However, in the context of Wasm binaries, these issues are often attributed to generic and repetitive instructions (e.g., load/store), many of which are subject to control and data dependency constraints. Even simple variable assignments transform into abstract operations involving storing and loading from memory. Consequently, merely detecting these instructions in Wasm binaries is impractical and prone to errors. Table 9 in Appendix presents a selection of sinks (error locations) for four types of memory bugs and six kinds of undefined behaviors, illustrating how they manifest in both source code and Wasm binaries.

To address this issue, we propose the *Wasm selective sanitization* mechanism, which filters the sinks requiring instrumentation by analyzing control and data dependencies for all anchor instructions. WBSan selects sanitization points by applying our defined static analysis passes and dynamically matching all instructions that could lead to issues. Next, we will sequentially introduce the dynamic matching patterns for various types of undefined behaviors and analysis passes for memory bugs. Table 3 lists the matching patterns and control and data dependency constraints we define for the six types of undefined behaviors.

**Shift-overflow**. Shift overflow happens when a bitwise shift operation exceeds the boundaries of the data type, potentially causing unexpected results or data loss. In Wasm binaries, The shift operation is represented by a direct instruction, shift, which pops two elements, a and b, from the stack and shifts b by a positions. Consequently, WBSan hooks all shift instructions in Wasm and associates them with their operands using local instructions.

**Integer-overflow**. In Wasm binaries, integer overflow bugs can arise in two scenarios: 1) Overflow of 32-bit and 64-bit integers,

represented by the basic Wasm variable types i32 and i64. 2) Overflow of 8-bit (char) and 16-bit (short) integers. For the first scenario, WBSan hooks the arithmetic operation instructions for 32-bit and 64-bit integers along with their operands, ensuring that the corresponding operands can be located in the *control&data dependency graph* for the instruction. For the second scenario, Wasm truncates the upper 24 or 16 bits of 32-bit integers through memory access and shifting, facilitating operations on 8-bit or 16-bit values. WBSan analyzes data dependencies to identify possible truncation or shifting operations, confirming that control flow paths support these operations. If WBSan finds that a 32-bit integer operation's data dependency leads to instructions like i32.store16 (store the lower 16 bits of a 32-bit value in memory) and a control flow path exists, it marks this as a sanitization point.

**Implicit-integer-sign-change.** Implicit integer sign change refers to the implicit conversion of integer types that can lead to unexpected results due to the change in sign. In Wasm, such sign conversions often occur during operations like i32.load8u, which read smaller bitwise numbers from larger memory units. Since smaller bit sizes may not be aligned with memory units, Wasm uses shift instructions afterward to clear the upper bits of the retrieved data, ensuring data cleanliness. WBSan hooks these five types of instructions (as shown in the Table 3) and records the data flow for subsequent instrumentation. Additionally, implicit integer sign changes can occur in function calls that use unsigned parameters, such as passing a negative number to the malloc function. WBSan marks such function calls as sanitization points.

**Null-dereference.** WBSan hooks all load instructions and their operands, and subsequently checks during instrumentation whether the addressed location is zero.

**Float-cast-overflow.** Flow casting overflow values in floating-point numbers occurs during type conversions , resulting in precision loss or overflow. The f32.demote_f64 instruction is used for converting double-precision floating-point numbers to single-precision in Wasm, and WBSan designates this instruction as a sanitization point. Additionally, i32.trunc_f32/f64 instructions to truncate the floating-point values to integers. Wasm programs check whether a floating-point number exceeds the range of integers; if it does, they directly replace its value with the minimum integer, which is -2,147,483,648 (I32_MIN). While this mechanism provides a form of mitigation, it merely replaces the value when an overflow occurs without accompanying any error messages. WBSan hooks this detection mechanism and marks the i32.trunc_f32 or i32.trunc_f64 instructions as sanitization points.

**Float-divide-by-zero.** Existing Wasm sandboxes report an error when encountering integer division by zero withour checking the float number. Therefore, WBSan hooks the f32.div and f64.div instructions as sanitization points to detect floating-point division by zero errors.

Unlike undefined behaviors, memory bugs often involve more intricate control and data dependencies. Although *Wasm shadow memory* can detect all memory bugs by hooking all memory access instructions. WBSan provides a set of passes for analyzing two types of memory bugs, *use-after-free* and *memory-leak* on the *control&data dependency graph*. WBSan hook only the suspicious load/store instructions after using defined analysis passes to reduce the overhead caused by instrumentation. Additionally, WBSan

**Figure 2: Instrumented Wasm with *shif-overflow* check.**

applies PCA [42], a static analysis tool for *memory—leak*, assessing whether the data has been properly released by analyzing whether memory objects have been freed through control and data dependency. WBSan designates the identified potential memory allocations and memory read/write instructions as sanitization points. Algorithm 1 and Algorithm 2 in Appendix outline the detailed analysis algorithms corresponding to *use-after-free* and *memory-leak*.

**3.2.4 Customized instrumentation.** WBSan will only perform instrumentation at the designated sanitization points after they have been identified. WBSan maps all identified sanitization points to their corresponding control and data dependencies in the target binary, allowing for the retrieval of the operands of anchor instructions. It then inserts the appropriate detection mechanisms at these points. We developed an instrumentation tool consisting of approximately 2K lines of Rust code, utilizing three Rust crates: Wasabi [37] (for analysis), WasmParser [5] (for parsing), and WasmEncoder [6] (for encoding and decoding Wasm). Figure 2 illustrates one instrumentation example. WBSan instruments instructions to check whether the operands are greater than 32 (Line 7-10) and whether both the operands and backup operands are positive numbers (Line 3-6 and Line 11-13). We have design unique instrumentation methods for undefined behaviors and memory bugs as detailed in Table 10 in the Appendix for conciseness.

## 4 Evaluation

### 4.1 Evaluation Target

To evaluate WBSan, we aim to address three research questions:

- **RQ1 - Effectiveness:** How effective is WBSan in detecting memory bugs and undefined behaviors compared to the existing Wasm binary checkers, as well as source-level Wasm and native binary sanitizers?
- **RQ2 - Applicability of Wasm fuzzing:** Can WBSan be integrated into existing Wasm fuzzing framework?
- **RQ3 - Performance overhead:** Is the amount of runtime overhead and code increase incurred by WBSan acceptable?

Table 4: CWE Descriptions.

| CWE(s) | Cases | Description |
|--------|-------|-------------|
| 680 | 288 + 114 | Shift-Overflow (SO) |
| 190, 191 | 4,184 + 4,216 | Integer-Overflow (IO) |
| 195 | 576 + 576 | Implicit-integer-Sign-Change (ISC) |
| 681 | 54 + 54 | Float-Cast-Overflow (FCO) |
| 476 | 283 + 283 | Null-Dereference (ND) |
| 369 | 450 + 450 | Float-Divide-by-Zero (FDZ) |
| **Total** | **5,835 + 5,803** | Undefined behaviors |
| 415, 416 | 1,228 + 1,228 | Use-After-Free (UAF) |
| 401 | 1,268 + 1,268 | Memory-Leak (ML) |
| 122 | 3,460 + 3,460 | Heap-buffer-Overflow (HO) |
| 121 | 2,824 + 2,824 | Stack-buffer-Overflow (SO) |
| **Total** | **8,780 + 8,780** | Memory bugs |

**Note:** Cases include the filtered malicious samples plus their corresponding benign samples.

Table 5: Target Real-World Programs.

| Name | Input | Version | LOC | Files |
|------|-------|---------|-----|-------|
| Abc2metx | Alembic | 1.6.1 | 4.63K | 6 |
| Libsndfile | Audio | 1.2.2 | 65.33K | 3 |
| Flite | Text | 2.2.0 | 583.38K | 208 |
| Flac | Flac | 1.3.2 | 59.25K | 90 |
| Libtomcrypt | Text | 1.18.2 | 74.39K | 2 |
| Http-parser | Text | 2.9.4 | 7.67K | 7 |
| Libpng | Png | 1.6.35 | 40.88K | 18 |
| Jbig2dec | Jbig | 1.4.0 | 15.97K | 24 |
| Libtiff | Tiff | 4.3.0 | 87.38K | 7 |
| Openjepg | Bmp/Png/... | 1.5.1 | 207.26K | 241 |
| Pdfresurrect | Pdf | 0.23 | 1.65K | 2 |

## 4.2 Experimental Setup

**4.2.1 Dataset** We employ the Juliet benchmark test suite from NIST [49] as our dataset for detecting memory bugs and undefined behaviors, which encompasses 118 CWE (Common Weakness Enumeration) types. As shown in Table 4, 13 out of 118 CWEs are selected, which correspond to the four types of memory bugs and six types of undefined behaviors with 29,198 test cases in total. To ensure that all test samples can reliably trigger the target issues, we removed non-determinism, which include: 1) Functions that use random numbers for branching. 2) Functions labeled as *good* but that still exhibit memory bugs or undefined behaviors.

As shown in Table 5, to test whether WBSan can effectively integrate with existing Wasm fuzzing framework, we select 11 real-world programs from recent work [23, 38, 41] and github [7–9, 12]. These projects encompass intended WebAssembly use cases such as data processing, media file handling, and cryptography.

**4.2.2 Environment** We use Clang 10.0.0, Emscripten 3.1.45 (for the Juliet test suite), and *WASI SDK* 20 (for real-world projects) as the backend compilers. We perform our experiments on Ubuntu 20.04 LTS with an Intel(R) Xeon(R) 2.00GHz CPU, and 128GB RAM.

## 4.3 RQ1: Effectiveness of Bug Detection

To comprehensively evaluate the detection capability of WBSan, we tested bug detection capability on 12 CWEs corresponding to 6 types of undefined behavior and 4 types of memory errors, and compared it WBSan with: 1) Two state-of-the-art Wasm binary bug checkers, Wasmati and Fuzzm-canary. 2) Three prevalent Native binary sanitizers, Valgrind, QASan and Retrowrite. 3) Two source-level Wasm sanitizers, ASAan and UBSan, provided by Emscripten. Table 6 presents the comparison results. Since the Juliet test suite provides both bug-triggering samples and benign samples for the same test case, **Cases** represents the total number of bad functions plus good functions. **FN** represents *False Negatives*, which are malicious samples incorrectly classified as benign, while **FP** represents *False Positives*, which are benign samples incorrectly identified as malicious.

**4.3.1 Compared with Wasm binary checkers.** We carefully select Wasm bug checkers capable of detecting issues in Wasm binaries, specifically Wasmati [20] and fuzzm-canary [38]. Wasmati is a static analysis tool for detecting Wasm bugs that utilize a code property graph to identify memory errors and undefined behaviors. Fuzzm-canary instrument Wasm binaries for detecting stack and heap buffer overflows. WBSan achieves superior results in the detection of six types of undefined behaviors with zero false negative rate (successfully detects all problematic samples) and only a 4.7% false postive rate. For four types of memory bugs, it exhibits a 16.8% false negative rate and a zero false postive rate. The detection results of Wasmati across all CWEs are not ideal, as the analysis passes provided by Wasmati are often limited to specific cases, such as fixed function names and calling methods. Compared to Fuzzm-canary, WBSan perfectly detects all heap-buffer-overflows (0 < 67.2%) because *Wasm shadow memory* can dynamically check the valid ranges of the stack or heap during execution, and it also achieves a significant lead in stack buffer overflows (49.2% < 67.2%).

**4.3.2 Compared with source-level Wasm sanitizers.** Due to information loss during the compilation process [23], source-level sanitizers leverage richer information (specific types, array or pointer boundary, etc.) compared to those operating directly on binaries. Despite getting less code information, WBSan still outperforms UBSan for undefined behaviors (0 vs. 0.3%), and achieves a detection rate comparable to ASan for memory bugs (16.8% vs. 10.8%). This limitation also makes it difficult to determine whether variables are signed in Wasm binaries and distinguish between array and pointer, resulting in WBSan encountering 6% false positives in *integer-overflow* and 49.2% false positives in *stack-buffer-overflow*.

UBSan cannot detect the precision loss when converting double precision floating-point numbers (double) to single precision floating-point numbers (float), and WBSan performs better in detecting *float-cast-overflow* (0 vs. 40.1%). Since ASAN requires specific exit points in the program, WBSan also achieves better results in detecting *memory-leak* bug (7.1% vs. 74.5%).

**4.3.3 Compared with native binary sanitizers.** We carefully chose state-of-the-art binary sanitizers, Valgrind [48], QASan [29] and Retrowrite [25] to compare with WBSan for detecting memory bugs. Because *Wasm shadow memory* can dynamically maintain the valid memory range at runtime by inserting red zones, WBSan

**Table 6: Memory bug and undefined behavior detection results on Juliet test suite compared to:1) Wasm binary checkers (Wasmati and Fuzzm-canary), 2) Source-level sanitizer (ASan and UBSan) and 3) Native binary sanitizers(Valgrind, QASan and Retrowrite).**

| CWE | WBSan | Wasmati | Fuzzm-canary | ASan | UBSan | Valgrind | QASan | Retrowrite |
|-----|-------|---------|--------------|------|-------|----------|-------|------------|
| | **FN/FP** | **FN/FP** | **FN/FP** | **FN/FP** | **FN/FP** | **FN/FP** | **FN/FP** | **FN/FP** |
| 680 (SO) | **0/0** | 288(*100%*)/0 | - | - | **0/0** | - | - | - |
| 190, 191 (IO) | **0**/254(*6.0%*) | 4,126(*98.6%*)/0 | - | - | **0/0** | - | - | - |
| 195 (ISC) | **0/0** | 560(*97.2%*)/0 | - | - | **0/0** | - | - | - |
| 681 (FCO) | **0/0** | 54(*100%*)/0 | - | - | 18(*40.1%*)*/0 | - | - | - |
| 476 (ND) | **0/0** | 283(*100%*)/0 | - | - | **0/0** | - | - | - |
| 369 (FDZ) | **0/0** | 438(*97.3%*)/0 | - | - | **0/0** | - | - | - |
| **Total** | **0**/254(*4.7%*) | 5,749(*98.5%*)/0 | - | - | 18(0.3%)/**0** | - | - | - |
| 415, 416 (UAF) | **0/0** | 1,194(*97.2%*)/0 | - | **0/0** | - | 22(*1.8%*)/0 | 82(*6.7%*)/0 | 337(*27.4%*)/0 |
| 401 (ML) | **90(*7.1%*)/0** | 1,268(*100%*)/0 | - | 945(*74.5%*)/0 | - | 698(*55.0%*)/0 | 698(*55.0%*)/0 | 266(*20.9%*)/0 |
| 122 (HO) | **0/0** | 3,400(*91.2%*)/76(*2.0%*) | 1,899(*67.2%*)/**0** | **0/0** | - | 1,997(*57.7%*)/**0** | 532(*15.4%*)/0 | 1,826(*52.8%*)/0 |
| 121 (SO) | 1,388(*49.2%*)/**0** | 2,520(*89.2%*)/144(*5.1%*) | 1,899(*67.2%*)/**0** | **0/0** | - | 2,107(*74.6%*)/**0** | **1,381(*48.9%*)/0** | 1,632(*57.8%*)/0 |
| **Total** | 1,478(*16.8%*)/**0** | 8,382(*95.5%*)/220(*2.5%*) | 3,798(*43.3%*)/**0** | 945(*10.8%*)/**0** | - | 4,824(*55.0%*)/**0** | 2,693(*30.7%*)/**0** | 4,061(*46.3%*)/**0** |

**Note:** The first six rows of CWEs belong to undefined behaviors, while the last four belong to memory bugs. ASan and UBSan are source-level sanitizers provided by Emscripten. * indicates UBSan can not detect double-to-float samples. **FN** indicates false negative and **FP** indicates false postive.

performs with a false negative rate of 0, 7.1%, and 0 for *use-after-free*, *memory-leak*, and *heap-buffer-overflow* respectively, which all surpass three native sanitizers. WBSan also shows comparable results in detecting *stack-buffer-overflow* among native binary sanitizers (49.2% for WBSan and 48.9% for QASan).

> **Answer:** WBSan is currently the most effective Wasm binaries bug detection tool for memory bugs (16.8% false negetive and 0 false postive) and undefined behaviors (0 false negetive and 4.7% false postive), achieving results comparable to source-level tools (despite with less code information). Compared to established binary sanitizers, WBSan achieves leading results in three of four types of memory bug.

## 4.4 RQ2: Combine with Wasm Binary Fuzzing

To answer RQ2, we integrat WBSan with a Wasm fuzzing framework fuzzm [38], and conduct 120 hours of fuzzing on 11 real-world programs. Fuzzm incorporates canary instrumentation that implements heap and stack canary checks by inserting random numbers at the memory boundaries, subsequently verifying these values during deallocating and function returns.

Table 7 presents the results of fuzzing process, detailing crashes detected, as well as the unique paths identified. Unique path refers to the number of paths maintained by fuzzer that can trigger higher code coverage; a larger value indicates greater code coverage achieved during the fuzzing process. WBSan detects more crashes than fuzzm (1,174 vs. 556) within the same duration of time. In five out of the eleven programs where fuzzm was able to find crashes, WBSan identifies a greater number of crashes (*abc2metx*, *libpng*, *pdfresurrect*, *openjepg*, *flite*). WBSan discovers 37 new crashes in *jbig2dec* where fuzzm did not find any crash. WBSan finds more unique paths than fuzzm (162,385 vs. 22,237) in programs where crashes were detected, indicating higher code coverage because WBSan can trigger more crashes, helping the fuzzer build a richer seed corpus and guiding fuzzing to explore deeper code space.

**Table 7: 120-hours Wasm binary-only fuzzing results on 11 real-world programs.**

| Real-world program | Crash | | Unique path | |
|--------------------|-------|-------|-------------|-------|
| | Fuzzm | WBSan | Fuzzm | WBSan |
| Abc2metx | 170 | 561 | 1,264 | 115,869 |
| Flac | 0 | 0 | 1,512 | 1,511 |
| Jbig2dec | 0 | 37 | 2,459 | 3,931 |
| Libpng | 118 | 135 | 761 | 853 |
| Libtiff | 0 | 0 | 1,727 | 1,785 |
| Pdfresurrect | 58 | 86 | 517 | 33,959 |
| Openjepg | 124 | 264 | 11,310 | 16,107 |
| Libsndfile | 0 | 0 | 152 | 152 |
| Libtomcrypt | 0 | 0 | 53 | 53 |
| Flite | 86 | 91 | 1,997 | 2,064 |
| Http-parser | 0 | 0 | 503 | 501 |
| **Total** | 556 | 1,174 | 22,237 | 162,385 |

**Note: Unique path** represent the path found that can trigger unexplored code, with a larger value indicating higher code coverage.

> **Answer:** WBSan can effectively integrates with existing Wasm binary fuzzing framework, enabling the detection of more crashes (1,174 > 556) and achieving higher code coverage (593.96% more unique paths).

## 4.5 RQ3: Performance Overhead

We assess the impact of WBSan on Wasm binary performance overhead through two metrics: the binary size increase and the runtime overhead. We obtain the binary size increase and runtime overhead of WBSan using a standalone Wasm benchmark designed for performance evaluation, Wasm-r3 [17]. We also compare the average execution speed of 11 real-world programs with source-level sanitizers UBSan and ASan.

**4.5.1 Binary size increase.** We evaluate the binary size increase on 25 Wasm binaries in the Wasm-r3 benchmark. WBSan instrumentation results in an average increase of 72.1% in the size of

**Table 8: Average Execution Time**

| Program | Average Execution Time (ms) | | | |
|---|---|---|---|---|
| | Original | WBSan | UBSan | ASAan |
| Abc2metx | 0.67 | 1.07(*1.59×*) | 0.78(*1.16×*) | 6.81(*10.1×*) |
| Flac | 2.13 | 3.98(*1.87×*) | 3.11(*1.46×*) | 12.84(*6.03×*) |
| Jbig2dec | 1.06 | 3.63(*3.44×*) | 3.02(*2.86×*) | 8.21(*7.76×*) |
| Libpng | 0.33 | 0.72(*2.21×*) | 0.65(*2.00×*) | 1.28(*3.90×*) |
| Libtiff | 16.96 | 33.11(*1.95×*) | 18.48(*1.09×*) | 37.15(*2.19×*) |
| Pdfresurrect | 37.60 | 52.29(*1.39×*) | 48.93(*1.30×*) | 561.23(*14.93×*) |
| Openjepg | 0.43 | 0.93(*2.16×*) | 0.67(*1.56×*) | 1.99(*4.63×*) |
| Libsndfile | 3.44 | 5.19(*1.51×*) | 4.93(*1.43×*) | 10.27(*2.99×*) |
| Libtomcrypt | 3.24 | 4.85(*1.50×*) | 6.47(*2.00×*) | 9.70(*2.99×*) |
| Flite | 111.17 | 186.96(*1.68×*) | 151.16(*1.36×*) | 274.47(*2.47×*) |
| Http-parser | 2.08 | 4.14(*1.99×*) | 3.24(*1.56×*) | 9.54(*4.59×*) |
| **Average** | 1× | 1.94× | 1.62× | 5.69× |

**Note:** × denotes the multiple of the runtime compared to the Original.

Wasm binaries, with a minimum increase of 50% and a maximum of 210%. Figure 3 in Appendix lists the specific code size growth.

**4.5.2 Runtime overhead.** Table 8 shows the average runtime of the original Wasm, WBSan, UBSan, and ASan after one thousand executions. Compared to uninstrumented Wasm binaries, the average runtime of the Wasm instrumented with WBSan is approximately 1.94 times greater. The additional runtime overhead incurred by WBSan is about 19.75% higher than that of UBSan and 65.91% lower than ASan. The overhead associated with ASan is significantly greater due to its extensive stack tracing [53]. Our analysis reveals that stack tracing in ASan accounts for nearly 80% of the runtime, as it provides detailed stack trace information through JavaScript or Host runtime APIs, even for code that does not contain issues.

> **Answer:** WBSan exhibits performance with acceptable file size increase (72.1%) and runtime overhead (1.94×) compared with compiled-based instrumentation (1.62× and 5.69× compared to uninstrumented).

## 5 Discussion

### 5.1 WebAssembly and Sanitizer

As a newly emerging language, WebAssembly presents challenges for analysis due to its different program architecture, and target platforms. Existing binary Wasm fuzzing methods lack of sanitization method for triggering different kinds of bugs. To address this, WBSan attempts to analyze and detect these bugs at the binary level. WBSan achieves a leading bug detection rate while maintaining acceptable overhead. WBSan can be used as a standalone sanitization tool for developers, or combined with fuzzing for more in-depth WebAssembly testing without source code.

### 5.2 Threats to Validity

**Internal validity.** Our results may not lead to our implications because we identify the anchor instructions by analyzing Wasm binaries compiled with Emscripten and *WASI SDK*, but this does not preclude the possibility of other anchor instructions.

**External validity.** Our results might not generalize to broader samples. Our findings may depend on testing samples from the curated Juliet test suite, which consists of small test cases. And a wider variety of test datasets may yield different results.

**Construct validity.** We may not detect all the memory bugs and undefined behaviors. Due to the irretrievable loss of source information, such as the distinction between signed and unsigned integers in Wasm, even though we have attempted to introduce *Wasm shadow memory* and use static analysis as mitigation.

### 5.3 Improvements and Future Work

WBSan can instrument Wasm binaries generated by the mainstream Emscripten and *WASI SDK* compilers. Our future work aims to enable WBSan to extract more program semantic information, allowing for more accurate memory boundaries to be obtained from binaries even when some compilation information is lost. The approaches used for recovering and extracting the semantic structure of Wasm in these studies are ideas that we can draw upon [28].

## 6 Related Work

**WebAssembly Analysis.** Several works analyze the correctness of the Wasm. Stiévenart et al. [56] manually analyze discrepancies between the native and Wasm binaries on the Juliet benchmark. Some work design dynamic analysis or symbolic execution frameworks for WebAssembly [33, 37]. Wasabi [37] is a browser-based dynamic analysis framework for Wasm binaries. Wasmati [20] and Wasma [19] provide different kinds of graphs for static analysis. Other works perform studies on real-world Wasm usage [34], on Wasm compiler bugs [51], and on inlining optimizations in Wasm [52].

**Sanitizer.** *1) Source/IR-level sanitizer.* ASAan [53] and MSan [55] are address sanitizers by instrumenting at IR-level and inserting red-zone checks. UBSan [15] detects various undefined behaviors by inserting corresponding check modes in the source code (AST). EffectiveSan [27] employs dynamically typed checks for memory bugs and undefined behaviors in C/C++. AddressWatcher [46] is a memory-leak bug checker by tagging and tracking execution path. *2) Binary sanitizer.* Valgrind [54] detects memory value errors using red-zone insertion methods. Undangle [21] uses taint tracking for memory bugs. RetroWrite [25] implements a binary address sanitizer through binary rewriting for static instrumentation. ZAFL [47] is an instrumentation framework that achieves compiler-level performance. MTSan [23] leverages ARM hardware features and binary rewriting to implement a high-performance memory sanitizer. KUBO [43] is a static undefined behavior detector for the Linux kernel. The above works are all focused on native programs and kernels.

## 7 Conclusion

This paper presents the first WebAssembly binary sanitizer, WBSan, designed for detecting memory bugs and undefined behaviors. WBSan shows a great bug detection rate on the benchmark, making it the most effective tool for bug detection for WebAssembly binaries to date. WBSan integrates well with existing WebAssembly binary fuzzing framework with acceptable performance overhead. WBSan can effectively aid developers in detecting WebAssembly issues and strengthening current binary WebAssembly fuzzing methods.

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

# Appendices

---
**Algorithm 1:** Selective analysis pass for Use-after-free

---
N = [] ;
**for** m *in* `malloc_functions` **do**
    **for** n1 *in* `cfg.get_descendants(m)` **do**
        **if** `n1.type == "Call"` *and* n1 *in* `free_functions` *and*
        `ddg.reaches(n1,m)` **then**
            | Insert n1 in N
        **end**
    **end**
**end**
**for** n2 *in* N **do**
    **for** n3 *in* `cfg.get_descendants(n2)` **do**
        **if** `n3.type == "Load"` *or* `n3.type == "Store"` **then**
            **if** `ddg.reaches(n3,n2)` **then**
                | **Potential Use-after-free found.**
            **end**
        **end**
    **end**
**end**

---

---
**Algorithm 2:** Selective analysis pass for Memory-leak

---
Flag = False ;
**for** m *in* `malloc_functions` **do**
    **for** n1 *in* `cfg.get_descendants(m)` **do**
        **if** `n1.type == "Call"` *and* n1 *in* `free_functions` *and*
        `ddg.reaches(n1,m)` **then**
            | Flag = True ;
        **end**
    **end**
**end**
**if** `Flag == False` **then**
    | **Potential memory-leak found.**
**end**

---

Function `get_descendants(node)` represents the set of all child nodes obtained from the corresponding graph. Function `reaches` is used to determine whether a path exists between two points. Notably, both Algorithm 1 and Algorithm 2 detect potentially problematic Wasm binaries, designating these problematic instructions as selective sanitization points for subsequent instrumentation.

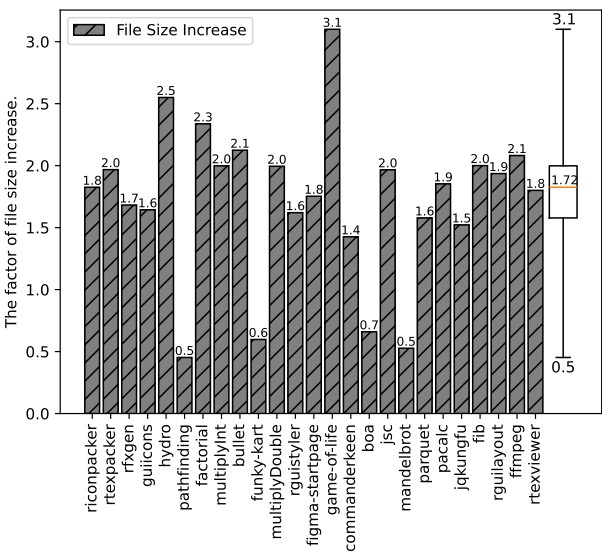

**Figure 3: Wasm file size increase on 25 test cases of Wasm-r3.**

---
**Algorithm 3:** Stack Instrumentation Procedure

---
**Input:** Wasm function F, Redzone size `RZ` (default: 16 bytes)
**Output:** Instrumented function F'
Insert at the beginning of F:
    $SP \leftarrow SP - RZ$
**foreach** *instruction* `instr` *in* F **do**
    **if** `instr` *is a return instruction* **then**
        Replace `instr` with:
            Unpoison(SP, RZ)
            SP ← SP + RZ; Execute `instr`
    **end**
**end**

**return** Instrumented function F'

---

---
**Algorithm 4:** Heap Instrumentation Procedure

---
**Function** `InstrumentedAlloc`(*size*)**:**
    memory ← OriginalAlloc(size + 32)
    preamble, userMemory, postamble ← memory, memory + 16, memory + size + 16
    Poison(preamble, 16); Poison(postamble, 16)
    Store size and magic value in preamble
    Unpoison(userMemory, size)
    **return** userMemory
**Function** `InstrumentedFree`(*ptr*)**:**
    preamble ← ptr - 16
    **if** *magic value in preamble is incorrect* **then**
        | Report double-free error
    **end**
    Unpoison(preamble, 16); Unpoison(ptr + size, 16)
    Poison(ptr, size) OriginalFree(preamble)
Replace `malloc`, `calloc` with `InstrumentedAlloc`; Replace `free` with `InstrumentedFree`

---

**Table 9: Source and Wasm binary sinks for memory bugs and undefined behaviors**

| Memory bug | Source sink (partial) | Wasm sink (partial) | Explanations |
|---|---|---|---|
| Heap-buffer-overflow | `int*buffer=new int[10];` `buffer[11]=1;` | `i32.store/i64.store` `i32.load/i64.load` | Writing or reading unallocated memory (heap) in Wasm. |
| Stack-buffer-overflow | `int buffer[10];` `buffer[11] = 1;` | `i32.store/i64.store` `i32.load/i64.load` | Writing or reading unallocated memory (stack) in Wasm. |
| Use-after-free | `data=malloc(...);` `free(data);` `printf("%s",data);` | `i32.store/i64.store/` `i32.load/i64.load` | Reading from or writing to deallocated memory objects. |
| Memory leak | `data=calloc(...);` `data=0;` | `i32.store/i64.store/` `i32.load/i64.load` | Pointer overwrite/missing deallocation. |

| Undefined behavior | Source sink (partial) | Wasm sink (partial) | Explanations |
|---|---|---|---|
| Shift-overflow | `data«100;` | `i32.shl/i64.shl` | The shift operand cannot exceed the bit width of the number being shifted and cannot be negative. |
| Integer-overflow | `result=data±INT_MAX;` | `i32.add/sub/mul/div/` `i64.add/sub/mul/div` | Arithmetic operations in Wasm have corresponding instructions. |
| Implicit-integer-sign-change | `int data = -1;` `malloc(data);` | `i32.load8u/i32.load16u/` `i64.load8u/load16u/load32u` | It typically occurs when retrieving a shorter unsigned number from 32-bit or 64-bit memory units. |
| Null-dereference | `int*data=NULL;*data=1;` | `i32.store/i64.store` | The dereferencing of NULL is reflected in the `store`. |
| Float-cast-overflow | `(float)doubleNumber;` ——— `(int)floatNumber;` | `f32.demoteF64` ——— `i32.trunc_f32/f64` | Double to float. Double or float to int. |
| Float-divide-by-zero | `(int)(100.0/0);` | `f32.div/f64.div` | Wasm hosts (such as browsers and runtimes) inherently check integer division by zero; here, we only detect floating-point division by zero. |

**Note: Source sink** indicates the point in the source code where issue occur. **Wasm sink** indicates the location where they occur in the Wasm binary.

**Table 10: Instrument logic for undefined behaviors**

| Undefined behavior | Matching pattern(s) | Instrumented logic | Description |
|---|---|---|---|
| Shift-overflow | ①`local.get a` ②`local.get b` ③`i32.shift` | $b \geq 0 \ and \ 0 \leq a \leq w$ | $w$ represents the size of the variable; an `i32` integer has a width of 32 bits. |
| Integer-overflow | ①`local.get a` ②`local.get b` ③`i32.add/sub/mul` | Add: $((a > 0 \wedge b > 0) \wedge a + b < 0) == 0$ Sub: $((a > 0 \wedge (-b) > 0) \wedge a + (-b) < 0) == 0$ Mul: $a \wedge (a * b) == b$ | We compile these logic checks into Wasm functions, and directly insert these validation functions during instrumentation. |
| Implicit-integer-sign-change | ①`i32.load8u` ②`local.set a` | $(a \ll 24) \gg 24 == a$ | Similarly, for the `i32.load16u`, the shift operation can be replaced with 16. |
| Float-cast-overflow | ①`local.get a` ②`f32.demote_f64` ③`local.set b` | $Promote(b) == a$ | Promote refers to the extension of a 32-bit floating-point number to a 64-bit representation, which is achieved using the `f64.promote_f32` in Wasm. |
| | `block` ①`i32.trunc_f32` ②`local.set a` `end` ③`i32.const I32_MIN` | $a == I32\_MIN$ or $a == I64\_MIN$ | Similarly for `i64.trunc_f64`. |
| Null-dereference | ①`local.get a` ②`local.get b` ③`i32.load/store` | $a == 0$ | The first parameter (a) of the `load/store` instruction indicates the target address and the second (b) specifies the value. |
| Float-divide-by-zero | ①`local.get a` ②`local.get b` ③`f32.div` | $a == 0$ | `f32.div` instruction performs the operation of b/a. |

**Note:** X in **Matching pattern(s)** represents any specific anchor instruction corresponding to a particular undefined behavior.

