# OpenReview forum: "WBSan: Webassembly Bug Detection for Sanitization and Binary-Only Fuzzing"
_ACM.org/TheWebConf/2025/Conference — WWW 2025 Poster_

### Official Review · Reviewer_tc9U · 2024-11-20

**Novelty:** 5
**Technical Quality:** 5

**Review:**

The paper introduces a WebAssembly (Wasm) binary sanitizer designed to detect memory bugs and undefined behaviors in Wasm binaries. This tool applies static analysis and binary instrumentation, leveraging Wasm shadow memory and unique detection patterns, demonstrating superior bug detection rates and improved code coverage in fuzzing applications.

Strengths:
Without source code.
Static analysis and custom instrumentation effectively target Wasm’s unique memory structure.
Demonstrates significant improvements in detection accuracy and reduced false positives over existing Wasm sanitizers and native tools.

Weaknesses:
The most recent method used for comparison is from 2022 (Wasmati). Any more recent papers can be compare?
Please ensure all references are correctly cited. In Table 1, KAsan [3], [3] seems a GitHub issue.

**Questions:**

See above

**Reviewer Confidence:**

3: The reviewer is confident but not certain that the evaluation is correct

**Scope:**

4: The work is relevant to the Web and to the track, and is of broad interest to the community

---

### Official Review · Reviewer_oqZn · 2024-11-21

**Novelty:** 1
**Technical Quality:** 1

**Review:**

The paper proposes a WebAssembly (Wasm) binary vulnerability detection tool named “WBSan,” claiming exceptional performance in detecting memory errors and undefined behaviors. While the work demonstrates a degree of innovation and technical depth, the following areas require further improvement and clarification. Specific details can be found in QUESTIONS.

**Questions:**

1.	The challenges in detection (Section 2.3) are not clearly articulated. Table 1 provides minimal insight, merely listing and categorizing existing methods without sufficient detail. As a result, the motivation for the proposed approach is unclear. The paper must specify how the methods listed in the table perform in detecting crashes, hangs, memory errors, and undefined behaviors. Furthermore, it is essential to explain, from a design perspective, the limitations of source-level or compilation-dependent methods in detection and whether a binary-only approach provides any novel capabilities. The “gap” necessitating a binary-only solution remains ambiguous, especially considering tools like ASan already demonstrate robust performance for memory error detection.

2.	The claim regarding limited access to source code when interacting with Wasm modules needs concrete evidence. This significantly impacts the perceived value of a binary-only detection mechanism. From a security standpoint, source-level detection with complete contextual information appears to be a more comprehensive approach for identifying potential vulnerabilities.

3.	The concepts of "dynamic memory boundary detection" and "Wasm shadow memory" are introduced, yet these techniques have been widely employed in traditional memory detection tools like ASan and Valgrind. The paper must clarify how these methods have been specifically adapted or enhanced for the Wasm architecture. For example, it should provide a detailed explanation of the mechanism for “red zone” insertion and its impact on Wasm-specific memory structures, such as linear memory and managed stacks. Additionally, while Table 3 lists six detection patterns for undefined behaviors, the descriptions of their implementation remain abstract. More detailed explanations, such as how the tool handles complex control and data dependencies, are necessary, including pseudocode or in-depth technical details.

4.	The paper utilizes the Juliet test suite, which covers vulnerabilities defined by the CWE standard. However, it is unclear whether the sample size and diversity are sufficient to represent the complexity of real-world scenarios.

5.	WBSan incurs a runtime overhead of 1.94x compared to uninstrumented Wasm code and an average binary size increase of 72.1% after instrumentation. These results necessitate a discussion of the tool’s usability on resource-constrained devices and whether the increased binary size poses challenges in low-bandwidth network environments, particularly concerning transmission and deployment.

6.	The paper states that WBSan incurs a false positive rate of 49.2% in detecting stack-buffer-overflows due to limited code information. Such a high false positive rate may severely undermine the utility and reliability of the tool in practical scenarios. Can the authors propose enhancements or modifications to their approach that could reduce this rate?

7.	The paper employs static analysis to identify "anchor instructions" for potential bug occurrences. Given the known limitations of static analysis, such as its inability to fully understand runtime context or dynamic data flows, there are concerns about the comprehensiveness and accuracy of this approach. How do the authors ensure that their method effectively captures relevant bugs without an extensive number of false negatives or overlooking complex bug patterns?

**Reviewer Confidence:**

4: The reviewer is certain that the evaluation is correct and very familiar with the relevant literature

**Scope:**

3: The work is somewhat relevant to the Web and to the track, and is of narrow interest to a sub-community

---

### Official Review · Reviewer_UWwK · 2024-11-23

**Novelty:** 5
**Technical Quality:** 5

**Review:**

This paper presents an innovative security analysis approach for WebAssembly, introducing the WBSan tool designed to detect memory bugs and undefined behaviors in Wasm binaries. By combining static analysis, binary instrumentation, and shadow memory techniques, the paper addresses challenges unique to the Wasm ecosystem.

Pros:
1. The paper is well-organized, following a clear progression from problem motivation to methodology, experiments, and conclusions. Core concepts, such as Wasm shadow memory and selective sanitization, are effectively explained.
2. The integration of shadow memory into Wasm binaries and its novel instrumentation techniques are a significant contribution.
3. The experimental evaluation is comprehensive, with detailed comparisons between WBSan and state-of-the-art tools across multiple benchmarks.

Cons:
1. Despite its novelty, the paper could provide more insight into how WBSan compares or contrasts with general binary sanitizers outside the Wasm domain, such as those for ELF or PE formats.
2. Certain technical terms, such as "anchor instructions" and "red zones," might require additional explanation to enhance accessibility for readers less familiar with WebAssembly's technical intricacies.
3. Although the experiments are thorough, they rely predominantly on synthetic datasets and curated benchmarks, limiting the exploration of real-world variability.
4. The tool’s practical usability in constrained environments may be affected by its reported binary size increase (72.1%) and runtime overhead (1.94×), which could pose challenges for large-scale or resource-limited deployments.

**Questions:**

1. How does the shadow memory mechanism in WBSan handle the challenges posed by increasingly large Wasm binaries, and are there any specific strategies employed to minimize memory consumption?
2. Can the techniques used in WBSan, such as shadow memory and binary-only sanitization, be applied to other binary formats or execution environments like ELF or PE? Additionally, how do these techniques compare to existing solutions in those contexts?
3. The selective sanitization method aims to reduce overhead. Could you provide detailed examples of how specific regions in the binary are prioritized, and what impact this has on the overall bug detection effectiveness?
4. Is there an ablation study that quantifies the contributions of individual components in WBSan, such as shadow memory and dependency analysis, to the tool's overall performance and accuracy?
5. WBSan is described as being integrated with fuzzing frameworks. Does this integration introduce any compatibility challenges with popular fuzzing tools such as AFL or libFuzzer, and if so, how are these issues addressed?

**Reviewer Confidence:**

3: The reviewer is confident but not certain that the evaluation is correct

**Scope:**

4: The work is relevant to the Web and to the track, and is of broad interest to the community

---

### Official Review · Reviewer_ujEK · 2024-12-02

**Novelty:** 6
**Technical Quality:** 6

**Review:**

This paper introduces a WebAssembly (WASM) binary sanitizer designed to detect memory bugs and undefined behaviors in WASM binaries without requiring source code. WBSan combines static analysis with binary instrumentation to identify problematic instructions and integrates a shadow memory mechanism for tracking valid memory boundaries. It implements custom instrumentation for various error types, ensuring minimal performance overhead while preserving stack and type correctness. Compared to existing tools, WBSan achieves higher bug detection rates, improved code coverage, and lower false positives.

Pros:
Overall, this paper is well-written. The evaluation section is robust and compares WBSan with existing tools, providing a clear understanding of its advantages, particularly in terms of bug detection and code coverage.

Cons:
My primary concern is that the paper mentions using Rust code for instrumentation but does not go into enough technical detail on how the instrumentation is performed at a detailed level. This could make it harder for other researchers to replicate the exact methodology. As the authors claim that the current binary rewriting techniques predominantly focus on specific instruction sets and lack a universally applicable framework for the WASM binaries, I would suggest the authors add transparency in implementation to allow easy replication and adaptation of the method.

**Questions:**

1) Can you provide the details of how WASM binaries are rewritten?
2) In your evaluation, WBSan was shown to find more crashes and achieve higher code coverage than existing fuzzing tools. However, could you elaborate on how WBSan influences the fuzzing process itself? Does it help guide the fuzzer more efficiently, or is the increase in crashes mainly due to its broader detection capabilities?

**Reviewer Confidence:**

4: The reviewer is certain that the evaluation is correct and very familiar with the relevant literature

**Scope:**

4: The work is relevant to the Web and to the track, and is of broad interest to the community

---

### Official Review · Reviewer_Arik · 2024-12-02

**Novelty:** 5
**Technical Quality:** 4

**Review:**

Thank you for submitting to WWW 2025! The paper was an interesting read on the current state of binary-only fuzzing in the context of Webassembly. Overall, I think the paper makes some valuable contributions and should be accepted eventually. However, there are some flaws in the evaluation that have to be addressed.

The paper's main contribution, a binary-only sanitizer for Webassembly that detects memory bugs and undefined behavior, is a valuable piece that was still missing from the Webassembly fuzzing ecosystem. It leverages some nice instrumentation for detecting memory bugs and undefined behavior. Although the instrumentation largely relies on known techniques, the evaluation underlines the validity of the approach.

The design of Wasm shadow memory is based on common sanitization techniques where red zones are inserted to detect violations. Initially, the implementation sounded very expensive, given that a potentially large amount of memory access instructions have to be hooked. This was, however, addressed by the evaluation, which showed an acceptable performance overhead. The implementation also requires instrumenting allocation and deallocation functions. This may be easy for standard targets compiled with Emscripten and alike, but how does it translate to third-party Webassembly? My guess would be that most programs have easy-to-identify allocation/deallocation function signatures. An evaluation of this would be nice.

The selective sanitization covers an appropriate amount of undefined behavior and can be extended if needed. Here, I have a question regarding the Null-dereference behavior. According to my understanding, a regular program may well use the memory at index 0. Figure 4 of [36] has a small overview of the memory layouts chosen by different compilers. For example, while clang does not allocate anything at address 0, emcc allocates the data section there. What determines if this is actually undefined behavior, and how does this reflect in false positives? According to Table 6 there are no false positives, but this then likely depends on the choice of compiler.

The research questions posed for the evaluation are interesting and match my expectations. The Juliet benchmark seems like an appropriate target to evaluate false positives and negatives. As stated before, this may, however, depend on the choice of compiler. Only using the WASI SDK here is fine in my opinion, but there should be some discussion on how the choice of compiler and backend may influence the results.

Now, let's move on to the observation that raised the most questions on my part. The evaluation of RQ2 claims that, on the one hand, WBSan allowed fuzzm to discover more crashes and, on the other hand, also led to greater code coverage. How can a sanitizer lead to an increase of several 1000% in coverage and 600% overall? From my understanding, it only allows for detecting violations. The magnitude of the increases especially intrigues me. While a better seed corpus can increase coverage, the magnitude indicates to me that the coverage was likely calculated on the entire binary, including the instrumentation. If this is not the case, this should be clarified. If the instrumentation was indeed included in the measurements, this part should be repeated. Instead, the coverage should be calculated on the original binary by, for example, taking the inputs generated by fuzzm+WBSan and testing them against the original binary. Also, I did not find any mention of the crashes being deduplicated. Without deduplication the increase may just be explained by different inputs that all trigger the same bugs. As such, the evaluation has to be repeated while accounting for duplicates and without measuring the instrumentation.

The choice of evaluation target is sensible, although there should be links to the respective projects. I was, for example, not able to find "abc2metx" as it returned zero search results on Google and Github.
The performance overhead, both regarding the binary size and execution time, is relatively low. Of course, the comparison with ASan is slightly skewed by the stack tracing. Is there a reason that your sanitizer does not implement stack tracing as well? Please also include the configuration of the different sanitizers used for benchmarking.
I am also not worried that no new vulnerabilities have been found, as all projects are regularly used as fuzzing benchmarks, and this paper only introduces a sanitizer, not a fuzzer.

Finally, since a big part of the motivation is the absence of source code and influence on the compilation process, it would also be nice to evaluate the fuzzm+WBSan combination of unknown targets to see if the fuzzing approach generalizes.
I do not see this as a necessity but rather a nice-to-have part of the evaluation, as it would also remove the bias introduced by the choice of compilers.

The writing quality is good, although there are occasional typos. It may be worth running everything through a spell checker to weed out the remaining typos.

Minor:
- "Daniel et al. [36]" should be "Lehmann et al. [36]"
- "Figure 2: Instrumented Wasm with shif-overflow check." is missing a 't'

## Pros

- valuable contribution
- sensible choice of evaluation targets

## Cons

- lack of evaluation on third-party WebAssembly
- unclear description of evaluation (crash deduplication and coverage measurements)

**Questions:**

- Please expand on how the evaluation was performed. Are the crashes deduplicated? Was the instrumentation also measured as part of the coverage?
- Why are the fuzzm sanitizers missing from Table 1?
- Is a Null-dereference always undefined behavior? Table 6 shows no false positives.
- What is "abc2metx"? Can I get a link to the project?

**Reviewer Confidence:**

2: The reviewer is willing to defend the evaluation, but it is likely that the reviewer did not understand parts of the paper

**Scope:**

3: The work is somewhat relevant to the Web and to the track, and is of narrow interest to a sub-community